# *EventEpi*—A natural language processing framework for event-based surveillance

**Auss Abbood**[1]*, **Alexander Ullrich**[1]*, **Rüdiger Busche**[2,3]*, **Stéphane Ghozzi**[1,4]*

**1** Robert Koch Institute (RKI), Berlin, Germany, **2** Osnabrück University, Osnabrück, Lower Saxony, Germany, **3** inserve GmbH, Hannover, Lower Saxony, Germany, **4** Helmholtz Centre for Infection Research (HZI), Brunswick, Lower Saxony, Germany

* abbooda@rki.de (AA); ullricha@rki.de (AU); rbusche@uos.de (RB); stephane.ghozzi@helmholtz-hzi.de (SG)

**Data Availability Statement:** Code and CSVs for figures in the manuscript are placed in https://github.com/aauss/EventEpi. The incident data base can be found at https://doi.org/10.6084/m9.figshare.12575978. The word embeddings trained

## Abstract

According to the World Health Organization (WHO), around 60% of all outbreaks are detected using informal sources. In many public health institutes, including the WHO and the Robert Koch Institute (RKI), dedicated groups of public health agents sift through numerous articles and newsletters to detect relevant events. This media screening is one important part of event-based surveillance (EBS). Reading the articles, discussing their relevance, and putting key information into a database is a time-consuming process. To support EBS, but also to gain insights into what makes an article and the event it describes relevant, we developed a natural language processing framework for automated information extraction and relevance scoring. First, we scraped relevant sources for EBS as done at the RKI (WHO Disease Outbreak News and ProMED) and automatically extracted the articles' key data: *disease*, *country*, *date*, and *confirmed-case count*. For this, we performed named entity recognition in two steps: EpiTator, an open-source epidemiological annotation tool, suggested many different possibilities for each. We extracted the key country and disease using a heuristic with good results. We trained a naive Bayes classifier to find the key date and confirmed-case count, using the RKI's EBS database as labels which performed modestly. Then, for relevance scoring, we defined two classes to which any article might belong: The article is *relevant* if it is in the EBS database and *irrelevant* otherwise. We compared the performance of different classifiers, using bag-of-words, document and word embeddings. The best classifier, a logistic regression, achieved a sensitivity of 0.82 and an index balanced accuracy of 0.61. Finally, we integrated these functionalities into a web application called *EventEpi* where relevant sources are automatically analyzed and put into a database. The user can also provide any URL or text, that will be analyzed in the same way and added to the database. Each of these steps could be improved, in particular with larger labeled datasets and fine-tuning of the learning algorithms. The overall framework, however, works already well and can be used in production, promising improvements in EBS. The source code and data are publicly available under open licenses.

for this manuscript can be found at https://doi.org/
10.6084/m9.figshare.12575966. This information
can be found in the manuscript aswell.

**Funding:** The project was funded by the German
Federal Ministry of Health through the Signale 2.0
project (www.rki.de./signale-project) with the
following funding code: ZMVI1-2517FSB418. The
funders had no role in study design, data collection
and analysis, decision to publish, or preparation of
the manuscript.

**Competing interests:** The authors have declared
that no competing interests exist.

## Author summary

Public health surveillance that uses official sources to detect important disease outbreaks
suffers from a time delay. Using unofficial sources, like websites, to detect rumors of dis-
ease outbreaks can offer a decisive temporal advantage. Due to the vast amount of infor-
mation on the web, public health agents are only capable to process a fraction of the
available information. Recent advances in natural language processing and deep learning
offer new opportunities to process large amounts of text with human-like understanding.
However, to the best of our knowledge, no open-source solutions using natural language
processing for public health surveillance exist. We extracted expert labels from a public
health unit that screens online resources every day to train various machine learning mod-
els and perform key information extraction as well as relevance scoring on epidemiologi-
cal texts. With the help of those expert labels, we scraped and annotated news articles to
create inputs for the machine learning models. The scraped texts were transformed into
word embeddings that were trained on 61,320 epidemiological articles and the Wikipedia
corpus (May 2020). We were able to extract key information from epidemiological texts
such as disease, outbreak country, cases counts, and the date of these counts. While dis-
ease and country could be extracted with high accuracy, date and count could still be
extracted with medium accuracy with the help of machine learning models. Furthermore,
our model could detect 82% of all relevant articles in an unseen test dataset. Both of these
functionalities were embedded into a web application. We present an open-source frame-
work that public health agents can use to include online sources into their screening rou-
tine. This can be of great help to existing and emerging public health institutions.
Although parts of the information extraction function robustly and the relevance scoring
could already save public health agent's time, methods to explain deep and machine learn-
ing models showed that the learned patterns are sometimes implausible. This could be
improved with more labeled data and optimization of the learning algorithms.

## Introduction

### Event-based surveillance

One of the major goals of *public health surveillance* is the timely detection and subsequent con-
tainment of infectious disease outbreaks to minimize health consequences and the burden to
the public health apparatus. Surveillance systems are an essential part of efficient early-warning
mechanisms [1, 2].

In traditional reporting systems the acquisition of this data is mostly a passive process and
follows routines established by the legislator and the public health institutes [2]. This process is
called *indicator-based surveillance*.

Hints of an outbreak, however, can also be detected through changed circumstances that
are known to favor outbreaks, e.g., warm weather might contribute to more salmonellosis out-
breaks [3] or a loss of proper sanitation might lead to cholera outbreaks [4]. Therefore, besides
traditional surveillance that typically relies on routine reporting from healthcare facilities, sec-
ondary data such as weather, attendance monitoring at schools and workplaces, social media,
and the web are also significant sources of information [2].

The monitoring of information generated outside the traditional reporting system and its
analysis is called *event-based surveillance* (EBS). EBS can greatly reduce the delay between the
occurrence and the detection of an event compared to IBS. It enables public health agents to

detect and report events before the recognition of human cases in the routine reporting system of the public health system [2]. Especially on the web, the topicality and quantity of data can be useful to detect even rumors of suspected outbreaks [5]. As a result, more than 60% of the initial outbreak reports refer to such informal sources [6].

Filtering this massive amount of data poses the difficulty of finding the right criteria for which information to consider and which to discard. This task is particularly difficult because it is important that the filter does not miss any important events (sensitivity) while being confident what to exclude (specificity). Without such a filter process, it is infeasible to perform EBS on larger data sources. Algorithms in the field of natural language processing are well suited to tap these informal resources and help to structure and filter this information automatically and systematically [7].

## Motivation and contribution

At the RKI, the *The Information Centre for International Health Protection* (*Informationsstelle für Internationalen Gesundheitsschutz*, INIG), among other units, performs EBS to identify events relevant to public health in Germany. Their routine tasks are defined in standard operating procedures (SOPs) and include reading online articles from a defined set of sources, evaluating them for their relevance, and then manually filling a spreadsheet with information from the relevant articles. This spreadsheet is INIG's EBS database, called *Incident Database* (*Ereignisdatenbank*, *IDB*). The existence of SOPs and the amount of time spent with manual information extraction and especially data entry lead to the idea to automate parts of the process.

Applying methods of natural language processing and machine learning to the IDB, we developed a pipeline that:

- automatically extracts key entities (disease, country, confirmed-case count, and date of the case count which are the mandatory entries of the IDB and thus are complete) from an epidemiological article and puts them in a database, making tedious data entry unnecessary;

- scores articles for relevance to allow the most important ones to be shown first;

- provides the results in a web service named *EventEpi* that can be integrated in EBS workflows.

We did not formally define what a "disease" was but rather followed the conventions at INIG. Although considering symptoms or syndromes might lead to earlier event detection, those were rarely entered in the IDB.

All code and data necessary to reproduce this work are freely available under open licenses: The source code can be found on GitHub under a GNU license [8], the IDB and word embeddings (see Training of the classifiers) on Figshare under a CC BY 4.0 license, [9] and [10] respectively.

## Related work

The Global Rapid Identification Tool System (GRITS) [11] by the EcoHealth Alliance is a web service that provides automatic analyses of epidemiological texts. It uses EpiTator [12] to extract crucial information about a text, such as dates or countries, and suggests the most likely disease the text is about. However GRITS cannot be automated and is not customizable. To use it in EBS, one would need to manually copy-paste both URLs and output of the analysis. Furthermore, GRITS does not filter texts for relevance but only extracts entities from provided texts.

The *recent disease incidents page* of MEDISYS [13, 14], which channels PULS [15, 16], tabularly presents automatically-extracted outbreak information from a vast amount of news sources. However, it is not clear how articles are filtered, how information is extracted, or how uncertain the output is. Therefore, it cannot be used as such and as it is a closed software we could not develop it further.

## Materials and methods

The approach presented here consists of two largely independent, but complementary parts: key information extraction and relevance scoring. Both approaches are integrated in a web application called *EventEpi*. After preprocessing the IDB, texts of articles from the RKI's main online sources have to be extracted. The full pipeline is shown in Fig 1. With the exception of the convolutional neural network (CNN) for which we used Keras [17], we used the Python package scikit-learn to implement the machine learning algorithms [18]. The exact configurations of all algorithms can be found in S1 Table.

### Key information extraction

Key information extraction from epidemiological articles was in part already solved by EpiTator. EpiTator is a Python library to extract named entities that are particularly relevant in the field of epidemiology, namely: disease, location, date, and count entities. EpiTator uses spaCy [19] in the background to preprocess text. One function of EpiTator is to return all entities of an entity class (e.g., disease) found in a text. However INIG, as other EBS groups, is mostly interested in the *key* entities of each class. Accordingly, the IDB contains a single value for each type of information. Thus, we needed to be able to filter the output of EpiTator to a single entity per class that best describes the corresponding event. In the case of the IDB these are *source*, *disease*, *country*, *confirmed-case count*, and the *date* of the number of confirmed cases of an outbreak article.

Before we could explore methods to find the aforementioned key entities, we applied standard cleaning and preprocessing to the IDB such that it could be fed into machine learning algorithms (see S1 Text). To find the key entities among those found by EpiTator, we explored two methods, a heuristic and classification-based approach which we refer to as key entity filtering (see Sec. Key entity filtering). If the filtered output of EpiTator for a given article matched the respective key information of the IDB, we then knew that the filter selected the correct key entities.

**Key entity filtering.**   A naive approach to finding the key entity out of all the entities returned by EpiTator is to pick the most frequent one (the mode). We call this the *most-frequent approach*. To find the key country, we focused only on the first three geographic entities mentioned in the text, since articles tend to contain mentions of other geographic entities different from the key country. To improve performance, we developed a learning-based approach for key date and confirmed-case count. This is shown in the *supervised learning* block in Fig 1.

For the learning approach, we took the texts of the articles published in 2018 from the two most relevant sources, WHO DONs [20] and ProMED [21, 22] (a list of the RKI's frequently used sources and the reason for selecting those two are described in S1 Text) and applied a sentence tokenizer using the Python library NLTK [23]. Tokenization is the process of splitting text into atomic units, typically sentences, words, or phrases.

We filtered all sentences to only keep those that contained some entity $e_{c,j}$ recognized by EpiTator, with $c$ being the class of the entity (date or confirmed-case count) and $j$ being the $j^{th}$ entity in a text. If an entity $e_{c,j}$ in a sentence matched the entry of class $c$ in the IDB, then we

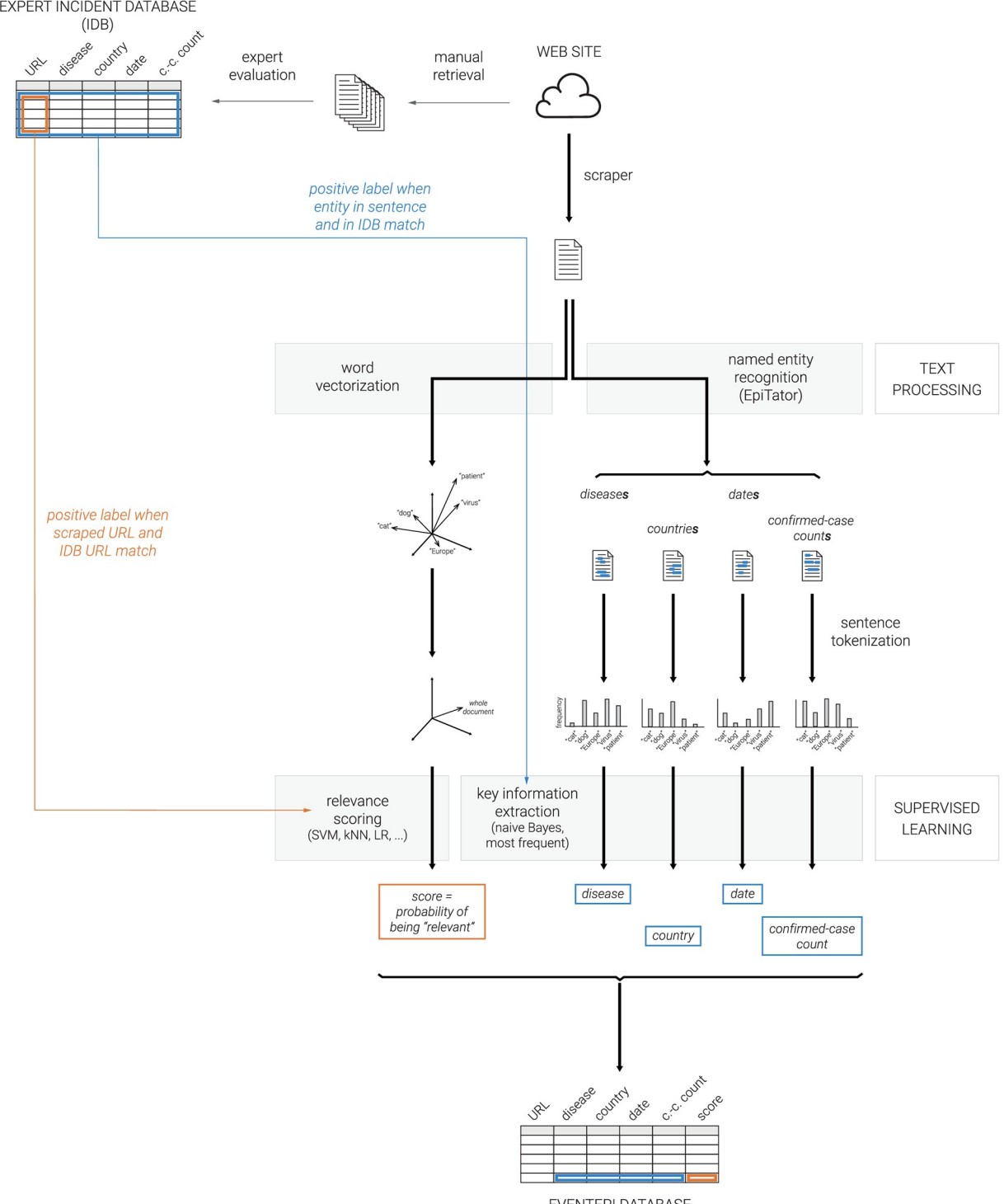

**Fig 1. An illustration of the *EventEpi* architecture.** The orange part of the plot describes the relevance scoring of epidemiological texts vectorized with word embeddings (created with word2vec), document embeddings (mean over word embeddings), and bag-of-words, and fed to different classification algorithms (support vector machine (SVM), k-nearest neighbor (kNN) and logistic regression (LR) among others). The part of *EventEpi* that extracts the key information is colored in blue. Key information extraction is trained on sentences containing named entities using a naive Bayes classifier or the most-frequent approach applied to the output of EpiTator, a epidemiological annotation software. The workflow ends with the results being saved into the *EventEpi* database that is embedded into *EventEpi*'s web application.

labeled this sentence as *key*. Every other sentence was labeled *not key*. The distribution of samples in the datasets obtained is summarized in S1 Fig.

Then we trained a Bernoulli naive Bayes classifier (Bernoulli NBC) [24] with these labeled sentences to learn the relevant properties of sentences that led to the inclusion of their information into the IDB. Before applying a classifier, a text needs to be represented as a vector of numbers (vectorization). During training, a Bernoulli NBC classifier receives for each input sentence a binary vector $b$ of the whole vocabulary (all the words seen during training) where the $i^{th}$ position of the vector indicates the $i^{th}$ term of the vocabulary. If the $i^{th}$ term $t_i$ is present in the input sentence, then $b_i = 1$ and 0 otherwise.

Based on the binary vectors and the corresponding labels, the Bernoulli NBC assigns probabilities to individual sentences of being *key* and *not key*. The key information for class $c$ was set to the entity recognized in the sentence that has the highest probability of being *key* and contains a recognized entity of class $c$. This method ensures that some entity is still chosen even if no sentence is being classified as *key*, i.e., if all sentences in a text have less than 50% probability of being *key*.

Additionally, we applied the multinomial NBC for comparison. The only difference to the Bernoulli NBC is that the multinomial NBC takes an occurrence vector $o$, with $o_i$ being the frequency of term $t_i$ in the text, as an input instead of a binary vector $b$. This approach is called bag-of-words. We combined bag-of-words with tf-idf (term frequency-inverse document frequency) where each term frequency is scaled so as to correct for overly frequent terms within and across documents. Formally, tf-idf is defined as

$$\text{tfidf}(t, d, D) = \frac{f_{t,d}}{\max\{f_{t',d} : t' \in d\}} \cdot \log \frac{N}{|\{d \in D : t \in d\}|}$$

where $t$ is a term from the bag-of-words, $d$ is a document, $D$ is the corpus of all epidemiological articles seen during training (containing $N$ documents) and $f_{t,d}$ is the frequency of term $t$ occurring in document $d$. The Bernoulli NBC might perform better on a small vocabulary, while the multinomial NBC usually performs equally well or even better on a large vocabulary [24]. We also applied further standard methods of text preprocessing (see S1 Text).

## Relevance scoring

The second part of developing a framework to support EBS was to estimate the relevance of epidemiological articles. We framed the relevance evaluation as a classification problem, where articles that were present in the IDB were labeled *relevant*. We had access to all assessments of the IDB of the year 2018 and therefore scraped all WHO DON and ProMED articles of the year 2018. This resulted in a dataset of 3236 articles, 164 of them labeled *relevant* and 3072 *irrelevant*. The exact statistics of the dataset are summarized in S2 Fig.

**Training of the classifiers.**   Modern text classifiers tend to use word embeddings [25, 26] for vectorization rather than the tf-idf and bag-of-words approach. Word embeddings are vector representations of words that are learned on large amounts of texts in an unsupervised manner. Proximity in the word embedding space tends to correspond to semantic similarity. This is accomplished by assigning similar embeddings to words appearing in similar contexts. First, we used standard pre-trained embeddings, trained on the Wikipedia 2014 and Gigaword 5th Edition corpora [27]. However, many terms specific to epidemiology were not represented. Thus, we produced custom 300-dimensional embeddings, training the word2vec algorithm [28] on the Wikipedia corpus of 2020 [29] and all available WHO DONs and ProMED Mail articles (61,320 articles). We applied the skip-gram approach and hierarchical softmax [28].

Those settings helped incorporating infrequent terms [30]. The embeddings were trained for five epochs. See S1 Text for information on computational resources and elapsed time.

Since we ultimately wanted to classify whether a whole document was *relevant*, we needed *document* embeddings. Although dedicated algorithms for document embeddings exist [31], we had not enough data to apply them meaningfully. However, taking the mean over all word embeddings of a document is a valid alternative [32] and suffices to show if learning the relevance of an article is possible.

A further issue was imbalance. Only a small fraction (5.0%) of the articles in the dataset was labeled *relevant*. Instead of discarding data from the majority class, we chose to up-sample the dataset using the ADASYN algorithm [33]. It generates new data points of the minority class by repeating the following steps until the proportion of minority and majority classes reaches the desired proportion (1:1):

1. choose a random data point $x_i$ (the document embedding of article $i$) of the minority class;

2. randomly choose another minority-class data point $x_{zi}$ among the 5-nearest neighbors of $x_i$;

3. generate a new data point $y_j$ at a random position between $x_i$ and $x_{zi}$ such that $y_j = ax_i + (1 - a)x_{zi}$ with $a$ drawn uniformly at random between 0 and 1.

One problem of up-sampling data is that it still uses the minority class to create new examples and this might hinder the generalizability of the classifier [34]. We used the imbalanced-learn package [35] to implement ADASYN. We compared different classifiers for the relevance scoring task using embeddings or the bag-of-words approach. Support vector machine (SVM), k-nearest-neighbors (kNN), logistic regression (LR) and multilayer perceptron (MLP) used document embeddings as features. The CNN operated on the word embeddings instead of the document embeddings. That way the striding filters of the CNN–if large enough–could learn relationships between adjacent words. We capped the input documents to a maximum of 400 words for the CNN. 597 documents contained less than 400 words which we filled up with zero embeddings such that each document has the same shape. For multinomial and complement NBCs, we used the bag-of-words approach since this feature representation coincides with the assumption of the NBC to predict a class given the occurrence (probability) of a feature. See S1 Table for a tabular, detailed view of the vectorizations and parameters used.

Finally, we used layer-wise relevance propagation [36] to make decisions of the CNN explainable. This is done by assessing which word embeddings passed through the CNN led to the final classification. We used iNNvestigate to implement this step [37].

## Evaluation

The output of the key entity filtering of texts (see Section Key entity filtering) was compared with the IDB entries of the respective texts. If a found key entity matched the IDB entry exactly we counted the filtered output as correctly classified. Less stringently, the extracted date was counted as correctly classified if it was within three days of the IDB entry. This is due to EpiTator's API which returns date ranges instead of single dates if parsing a text. A range of three days allows us to rightly overlay with EpiTator's date ranges. The performances of all classifiers are evaluated on a test set which consists of 25% of the whole dataset. We applied stratification to ensure that both classes are evenly distributed on the train and test set. The data for the CNN was split into a training (60%), validation (20%), and testing (20%) set with slightly different class composition due to different sampling strategies (see S1 Text for details). We consider a number of scores defined as functions of true positives (TP), true negatives (TN), false positives (FP) and false negatives (FN): precision = TP/(TP + FP); sensitivity = TP/(TP + FN);

specificity = TN/(TN + FP) and F1 = 2 · TP/(2 · TP + FP + FN) [38]. Since public health agents are interested in not missing any positives by classifying them incorrectly as negatives, we considered the sensitivity as a good indicator for the performance of the classifiers. As a measure for the overall accuracy, we preferred the index-balanced accuracy (IBA) [39], which has been developed to gauge classification performance on imbalanced datasets. It is based on sensitivity, i.e., the fraction of correctly classified relevant articles or key entities, and specificity, i.e., the fraction of correctly classified irrelevant articles or non key entities. It is defined as

$$IBA_\alpha = (1 + \alpha \cdot (\text{sensitivity} - \text{specificity})) \cdot \text{sensitivity} \cdot \text{specificity} \qquad (1)$$

where sensitivity − specificity is called the dominance and $0 \leq \alpha \leq 1$ is a weighting factor that can be fine-tuned based on how significant the dominating class is supposed to be. IBA measures the trade-off between global performance and a signed index that accounts for imbalanced predictions. It favors classifiers with better true positive rates, assuming that correct predictions on the positive class are more important than true negative predictions. As in the original publication [34], we use $\alpha = 0.1$.

## Results

In this section we present the performance of a series of key information extraction and relevance-scoring algorithms, and describe how the findings were embedded into the web application *EventEpi*.

### Performance of key date and count extraction

We identified the most probable true entity among the many proposed by EpiTator using the most-frequent approach and two NBCs. The most frequent approach worked well for detecting the key country (85% correctly classified) and disease (88% correctly classified). Note that EpiTator systematically failed to detect Nipah virus infection and anthrax. However, performing key information extraction of date and case-count entities using the most-frequent approach did not work and no entity was correctly classified (0% correctly classified in both cases).

   The performance of both NBC algorithms applied to extract key date and key confirmed-case count are shown in Tables 1 and 2 respectively. Confusion matrices and ROC curves present the performances in greater detail (see S5 and S7 Figs respectively).

   For both date and count information extraction, the Bernoulli NBC had the highest IBA and sensitivity. Thus, without offering perfect results, applying classification to the output of EpiTator enables key entity extraction for *date* and *confirmed-case count*. We might be able to improve the performance by hyperparameter tuning, or better feature extraction (e.g., looking for key words such as "confirmed"). Increasing the amount of training data however would probably not lead to much improvement (see S3 Fig).

**Table 1. Evaluation of the key *date* extraction.**

|  | Pre. | Sen. | Spec. | F1 | IBA | *key* sample size | *not key* sample size |
|---|---|---|---|---|---|---|---|
| **Multinomial naive Bayes** | **0.55** | 0.78 | **0.69** | 0.65 | 0.54 | 27 | 54 |
| **Bernoulli naive Bayes** | **0.55** | **0.81** | 0.67 | **0.66** | **0.55** | 27 | 54 |

For each classifier and label, the precision (Pre.), sensitivity (Sen.), specificity (Spec.), F1, index balanced accuracy (IBA) with $\alpha = 0.1$, and sample size for both classes, *key* and *not key*, of the test set is given. The best values for each score highlighted in bold.

**Table 2. Evaluation of the key *confirmed-case count* extraction.**

|  | Pre. | Sen. | Spec. | F1 | IBA | *key* sample size | *not key* sample size |
|---|---|---|---|---|---|---|---|
| **Multinomial naive Bayes** | **0.39** | 0.45 | **0.93** | **0.42** | 0.40 | 89 | 874 |
| **Bernoulli naive Bayes** | 0.20 | **0.81** | 0.67 | 0.32 | **0.55** | 89 | 874 |

Definitions and parameters are the same as in Table 1. The best values for each score highlighted in bold.

## Performance of relevance scoring

The results of the relevance scoring are shown in Table 3. The confusion matrices for these results are displayed in S6 Fig and the respective ROC curves in S8 Fig. A comparison on classifier trained on non up-sampled data showed no better performance than sensitivity of 0.03 and IBA of 0.02 (see S2 Table). While the logistic regression has the highest sensitivity (0.82) and IBA (0.61), no model has a higher precision than 0.22 which suggests that all classifier overfit the positive class.

The multinomial NBCs had a better performance (sensitivity and IBA) than the complement NBC, possibly because of the dataset imbalance [40]. Although the scores are relatively high in general, all classifier overfit the positive class. Overfitting usually can be avoided for some models. E.g, for the CNN, we can apply further dropout (random removal of nodes in the network during training time to minimize highly specified nodes), regularization (e.g., L2 to punish strong weighting of nodes), and early stopping (to minimize the difference of losses between the test and validation set). Most models can incorporate a class weight variable adjusted to control overfitting. Also, we did not optimize the decision boundary of the tested classifiers which, however, might improve the balance between both classes. All these points which fall in the task of hyperparameter optimization can be tackled in a separate step.

It is nevertheless interesting to use the CNN as an example for explaining what contributed to the classification. A plot of a layer-wise relevance propagation shows one example where a relevant article was correctly classified (Fig 2). We see that words like *500* in the beginning of the text are highlighted as strongly important for the classification of the text as being *relevant*. Also, the word *schistosomiasis*–an infectious disease caused by flatworms–is labeled as strongly relevant for the classification. Interestingly, it is also relevant for the classifier that this disease is treated with antiparasitic drugs (*anthelmintic*). Both make sense, since a very high number of cases of a dangerous infectious disease are of interest for public health agents. All other case numbers are labeled as slightly irrelevant which does not necessarily make sense. An event

**Table 3. The performance evaluation of the relevance classification.**

|  | Pre. | Sen. | Spec. | F1 | IBA | *relevant* sample size | *irrelevant* sample size |
|---|---|---|---|---|---|---|---|
| **Multinomial naive Bayes** | **0.22** | 0.42 | **0.93** | **0.29** | 0.37 | 38 | 771 |
| **Complement naive Bayes** | 0.19 | 0.61 | 0.87 | **0.29** | 0.51 | 38 | 771 |
| **Logistic regression** | 0.14 | **0.82** | 0.75 | 0.24 | **0.61** | 38 | 771 |
| **k-nearest neighbor classifier** | 0.12 | 0.63 | 0.77 | 0.20 | 0.48 | 38 | 771 |
| **Support vector machine** | 0.13 | 0.79 | 0.74 | 0.22 | 0.59 | 38 | 771 |
| **Multilayer perceptron** | **0.22** | 0.42 | **0.93** | **0.29** | 0.37 | 38 | 771 |
| **Convolutional neural network** | 0.14 | 0.55 | 0.78 | 0.23 | 0.42 | 42 | 606 |

For each classifier and label, the precision (Pre.), sensitivity (Sen.), specificity (Spec.), F1, index balanced accuracy (IBA) with $\alpha = 0.1$, and sample size of both classes, *relevant* and *irrelevant* articles, of the test set is given. The best values for each score are highlighted in bold.

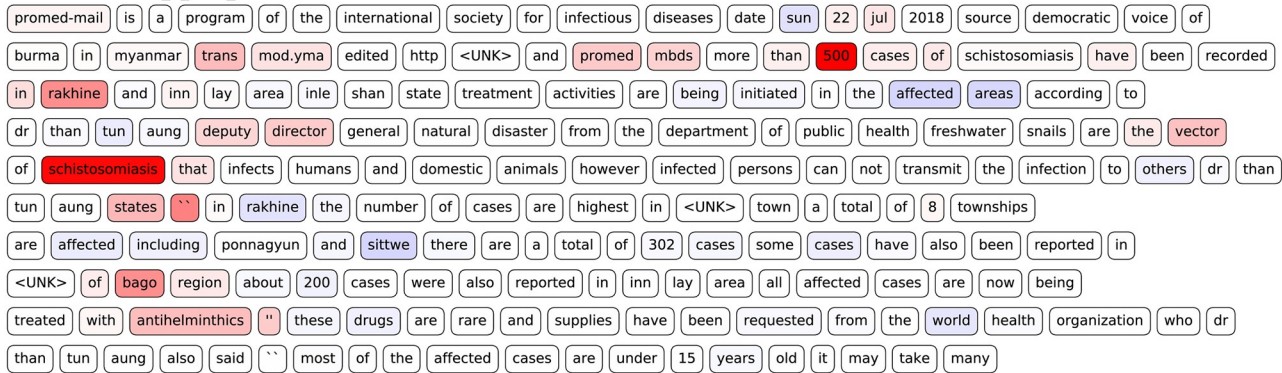

**Fig 2. A layer-wise relevance propagation of the CNN for relevance classification.** This text was correctly classified as relevant. Words that are highlighted in red contributed to the classification of the article being *relevant* and blue words contradicted this classification. The saturation of the color indicates the strength of which the single words contributed to the classification. <UNK> indicates a token for which no word embedding is available.

might be less relevant when out of 500 confirmed cases of some infectious disease half of the patients are in treatment.

The focus of this work was to show a proof of concept that classification methods can serve in determining the relevance of an article. We did not try to fine-tune all of the compared classifiers. Since training of the algorithms was only a matter of minutes, it might be cheap to perform hyperparameter optimization. Computational time and resources to train all models are described in S1 Text. For now, logistic regression (LR), although having a low precision, is preferred due to its good sensitivity and IBA. Although the relevance classification has not a very strong performance, it could already aid public health agents. The algorithms could be retrained every time articles are entered into the IDB to increase performance continuously. Indeed, testing the classifiers on fractions of the data shows a positive trend of performance (IBA) with increasing data size (see S4 Fig). Until very high performance can be achieved, relevance scores could be displayed and used to sort articles, but not to filter content.

## Web service

To showcase the analyses presented above and show how key information and relevance scoring can be used simultaneously to aid EBS, we developed the web application *EventEpi*. Fig 3 shows a screenshot of its user interface. *EventEpi* is a Flask [41] app that uses DataTables [42] as an interface to its database. *EventEpi* lets users paste URLs and automatically analyze texts from sources they trust or are interested in. The last step in Fig 1 shows how the *EventEpi* database is filled with the output of the key information extraction and relevance scoring algorithms. With our colleagues at INIG in mind, we integrated a mechanism that would automatically download and analyze the newest unseen articles from WHO DONs and ProMED. Currently, this process is slow and depends on pre-analyses for a good user experience. To allow for the integration of the functionality into another application, we also wrote an ASP.NET Core application to analyze texts via API calls.

## Conclusion

We have shown that novel natural language processing methodology can be applied in combination with available resources, in this case the IDB of the RKI, to improve public health surveillance. Even with limited datasets, EBS can be supported by automatic processes, such as

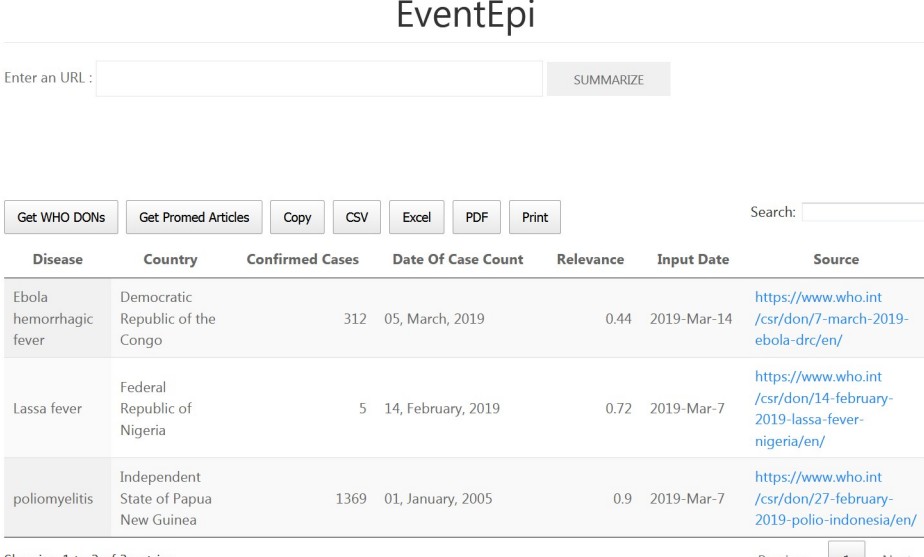

**Fig 3. A screenshot of the *EventEpi* web application.** The top input text field receives an URL. This URL is summarized if the `SUMMARIZE` button is pushed. The result of this summary is entered into the datatable, which is displayed as a table. The buttons `Get WHO DONs` and `Get Promed Articles` automatically scrape the last articles form both platforms that are not yet in the datatable. Furthermore, the user can search for words in the search text field and download the datatable as CSV, Excel or PDF.

pre-screening large amounts of news articles to forward a condensed batch of articles for manual review.

More work is needed to bring *EventEpi* into production. While key disease and country can satisfactorily be extracted, the performance of key date and confirmed-case count extractions needs to be improved.

Relevance scoring shows promising results. We believe it could already be helpful to public health agents, and could greatly be improved with fine-tuning and larger datasets.

The web application *EventEpi* is a scalable tool. Thus, the scope of EBS might be increased without comparable increase in effort. This is particularly relevant with the availability of automatic translation (for example DeepL [43]). It could allow an EBS team to access much more sources than those in the few languages its members typically speak without being overwhelmed. It is possible to provide better classifications that work for different languages using multilingual word embeddings [44], or a better key information extraction using contextual embeddings [45, 46] which adjust the embedding based on the textual context. Contrary to the relevance of a document, key information is mostly defined by its nearby words.

The same fundamental issues encountered in using machine learning in general apply here as well, in particular bias and explainability. Tackling individual biases and personal preferences during labeling by experts is essential to continue this project and make it safe to use. It will also be important to show *why EventEpi* extracted certain information or computed a relevance for it to be adopted but also critically assessed by public health agents. For artificial neural networks, we showed that layer-wise relevance propagation can be used in the domain of epidemiological texts to make a classifier explainable. For other models, model agnostic methods [47, 48] could be applied analogously.

At the moment *EventEpi* only presents results to the user. However it could be expanded to be a general interface to an event database and allow public health agents to note which articles

were indeed relevant as well as correct key information. This process would allow more people to label articles and thus expand the datasets, as well as help better train the relevance-scoring algorithms, an approach called active-learning [49].

With a large labeled dataset, a neural network could be (re)trained for the relevance classification. Later, transfer learning (tuning of the last layer of the network) could be used to adapt the relevance classification to single user preferences.

This work demonstrates how machine learning methods can be applied meaningfully in public health using data readily available: As experts evaluate and document events as part of their daily work, valuable labeled datasets are routinely produced. If systematically gathered and cataloged, these offer immense potential for the development of artificial intelligence in public health.

## Supporting information

**S1 Text.**
(PDF)

**S1 Table. Hyperparameter settings of the classifaction algorithms.** This tables lists the parameters and the vectorization methods stratified for the task and used models (naive Bayes classifier (NBC), support vector machine (SVM), k-nearest neighbors (kNN), logistic regression (LR), multi layer perceptron (MLP), and convolutional neural network (CNN)). More information on the used parameters can be found at https://scikit-learn.org.
(PDF)

**S2 Table. Performance evaluation of the relevance classification without up-sampling using ADASYN.** For each classifier and label, the precision (Pre.), sensitivity (Sen.), specificity (Spec.), F1, index balanced accuracy (IBA) with $\alpha = 0.1$, and sample size for both classes, *relevant* and *irrelevant* articles, of the test set is given. The best values for each score are highlighted in bold.
(PDF)

**S1 Fig. Sample distribution for key information extraction.** The number of articles used for each class (positive/negative, i.e. key/not key) for the partions of the dataset (train, test) are shown for each task.
(PDF)

**S2 Fig. Sample distribution for relevance scoring.** The number of articles used for each class (positive/negative, i.e. relevant/irrelevant) for the partions of the dataset (train, test, and for CNN validation) are shown for each task.
(PDF)

**S3 Fig. Learning curves for key count and date entity extraction.** Dependency of *key* (date and count) classifcation performance on training data size as measured using 5-fold cross validation for the multinomial and Bernoulli naive Bayes classifiers. The performance is measured by the IBA score. The points show mean scores, the shaded regions show the mean plus and minus one standard deviation on the cross validation folds.
(PNG)

**S4 Fig. Learning curves for relevance scoring.** Dependency of *relevance* classifcation performance on training data size as measured using 5-fold cross validation for different classifiers. The performance is measured by the IBA score. The points show mean scores, the shaded

regions show the mean plus and minus one standard deviation on the cross validation folds.
(PNG)

**S5 Fig. Confusion matrices of the key count and date entity extraction.** The plot shows the true and predicted labels of the test test in the key entitiy extraction task. The plots are stratified by algorithm (multinomial and Bernoulli naive Bayes classifier (NBC)) and task (key count and date extraction). Furthermore, the proportion of missclassified is shown below.
(PNG)

**S6 Fig. Confusion matrices of the relevance scoring.** The plot shows the true and predicted labels of the relevance scoring task. The plots are stratified by algorithm. Furthermore, the proportion of missclassified is shown below.
(PNG)

**S7 Fig. Receiver operating characteristics of the key count and date entity extraction.** The plot shows the true positive rate against the false positive rate stratified by algorithm (multinomial and Bernoulli naive Bayes classifier (NBC)) and task (key count and date extraction) and the area under the curve (AUC). The black, dotted middle shows the expected curve for random classifcation.
(PDF)

**S8 Fig. Receiver operating characteristics of the relevance scoring.** The plot shows the true positive rate against the false positive rate stratified by algorithm (complement naive Bayes classifier (compl. NBC), k-nearest neighbors (kNN), logistic regression (LR), multi layer perceptron (MLP), multi. NBC (multinomial naive Bayes classifier), support vecotor machine (SVM), and convolution neural network (CNN)) and the area under the curve (AUC). The black, dotted middle line shows the expected curve for random classifcation.
(PDF)

## Acknowledgments

We would like to thank Maria an der Heiden, Sandra Beermann, Sarah Esquevin, Raskit Lachmann and Nadine Zeitlmann for helping us on questions regarding epidemic intelligence, for providing us with data and for critical comments on the manuscript. We also thank Katarina Birghan for helping us using Wikidata and Fabian Eckelmann for his support in developing the *EventEpi* web application.

## Author Contributions

**Conceptualization:** Auss Abbood, Alexander Ullrich, Stéphane Ghozzi.

**Data curation:** Auss Abbood.

**Formal analysis:** Auss Abbood.

**Funding acquisition:** Alexander Ullrich, Stéphane Ghozzi.

**Investigation:** Auss Abbood.

**Methodology:** Auss Abbood, Rüdiger Busche.

**Project administration:** Alexander Ullrich, Stéphane Ghozzi.

**Software:** Auss Abbood.

**Supervision:** Alexander Ullrich, Stéphane Ghozzi.

**Validation:** Auss Abbood, Alexander Ullrich, Rüdiger Busche, Stéphane Ghozzi.

**Visualization:** Auss Abbood, Rüdiger Busche, Stéphane Ghozzi.

**Writing – original draft:** Auss Abbood.

**Writing – review & editing:** Alexander Ullrich, Rüdiger Busche, Stéphane Ghozzi.

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
