## [Decision Letter · Decision Letter 0]

29 Apr 2020

Dear Mr. Abbood,

Thank you very much for submitting your manuscript "EventEpi–A Natural Language Processing Framework for Event-Based Surveillance" for consideration at PLOS Computational Biology.

As with all papers reviewed by the journal, your manuscript was reviewed by members of the editorial board and by several independent reviewers. In light of the reviews (below this email), we would like to invite the resubmission of a significantly-revised version that takes into account the reviewers' comments.

We cannot make any decision about publication until we have seen the revised manuscript and your response to the reviewers' comments. Your revised manuscript is also likely to be sent to reviewers for further evaluation.

Sincerely,

Benjamin Althouse

Associate Editor

PLOS Computational Biology

Virginia Pitzer

Deputy Editor

PLOS Computational Biology

Reviewer's Responses to Questions

**Comments to the Authors:**

Reviewer #1: See attached.

Reviewer #2: Dear Authors,

Congratulations for your hard work.

It is well written. Additional information should be provided to help public health experts not familiar with natural processing language algorithms be able to judge the presented results.

From a public health point of view I have the following comments, questions and proposed modifications to the text:

General:

• Please use “Public health surveillance” instead of “Infectious disease epidemiology” or “epidemiological surveillance” in the context of this paper.

• Avoid the use of NLP acronym, consider using the full term “natural language processing” across the text?

• Please define each acronym at least once (for example TPR: true predictive ratio).

• Specify what you mean by “disease”. Is this limited to laboratory specific diseases such as measles, cholera or yellow fever; or it also includes syndromes such as cutaneous rash, watery diarrhoea, jaundice? This is especially important for EBS, as its purpose is mainly to detect unknown or unexpected diseases that cannot be well captured by the routine data reporting performed by healthcare facilities.

Author summary:

• What did this research do and find: 4th point (last): misleading sentence, as mentioned in the results only countries and diseases were correctly detected, not dates or counts.

Introduction

• Line 3 – 4: One of the most important goals of "Public health surveillance" is the "timely" detection and response to an acute public health event; the other being to monitor the health status of the population to drive health policy. In this paper, early detection and response is the topic of interest, yet public health surveillance cannot be reduced to that.

• Lines 14-15: “traditional surveillance” relies on “routine reporting from healthcare facilities” and not from “laboratory confirmation” (most cases are not laboratory confirmed in many settings and for many diseases).

• Lines 20-22: add “routine” in the sentence: “It enables public health agents […] recognition of human cases in the **routine** reporting system”.

• Line 29: why the use of the word “precision” instead of “specificity”?

• Lines 45-46: why only “confirmed-case count”? And not counts of suspect cases, for example?

Figure 1

• Don't use the light "pink-orange" background for boxes as it is confusing with the orange part of the figure describing the relevance scoring.

• Define briefly "word2vec", "EpiTator", "SVM", "kNN", "LR" either in the figure or in its description.

• To facilitate understanding, maybe consider splitting the "supervised learning" row in two: the classification phase (relevance scoring), and the identification of the appropriate data (key information extraction). And use the terms "relevance scoring" and "key information extraction" in the figure.

Material and methods:

• Add a section to detail how you assessed the performance of the key information extraction and of the classifiers, this would encompass among other lines 85-89, 267-273 in the methods, and some paragraphs from the results, for example lines 288 to 295.

• In the above proposed section, please add a quick description of all indicators used to assess each extraction and classification method (i.e. indicators described in the results and in tables 2 to 4).

• For epidemiologists the term “sensitivity” will be clearer than “recall”.

• Lines 79-80: if correct, add “in each class” in the following sentence: “However INIG, as other […] in the keys entitities *in each class*”.

• Lines 120-121: you mention the problem of events/incidents involving several countries but you don’t specify how you solved it (for example to have an accurate number of cases for each country). Please specify how many events/incidents it concerned and how you solved this problem. Why didn’t you remove these events for training the algorithm?

• Table 1: please specify what a “sample” is; and what is considered “positive” or “negative” samples.

• Lines 159-161: if I understood well, the key information for a single class is the recognized entity from the sentence with the highest probability. How did you deal with sentences having more than one recognized entity for a single class?

• Please make clear the number of articles used in your samples:

◦ As I understand, you had 3232 articles, 160 relevant (included in the IDB) and 3072 irrelevant.

◦ And then you tested your classifiers with 20% of your sample, please provide figures of relevant and irrelevant articles used to test your classifiers.

◦ As I understand, you had two classes: relevant or irrelevant article. If I am correct, please make it clear, including in the sections related to the test of the classifiers.

Results:

• When you say all countries and diseases (except one) were correctly recognized, please provide the figures; i.e. for how many articles presents in the IDB country and disease were correctly extracted. Same for dates and counts.

• Lines 299-302: not clear what the results related to EpiTator and the ones related to the most frequent approach are.

• Please also provide figures for table 4. For example, for each classification modality, how many documents were classified as relevant and not relevant, how many were truly relevant and how many truly not relevant?

• Similarly, it would be good to know if other incidents not logged in the IDB but still of interest were identified using the automatized screening approach, i.e. documents identified as “relevant” by the classifier, that were not in the IDB, but that should have been after a review by a public health expert.

Conclusion:

• The most important added value of the tool would be to pre-screen large amounts of data to identify a sample that would be then manually screened by public health experts.

**Have all data underlying the figures and results presented in the manuscript been provided?**

Reviewer #1: Yes

Reviewer #2: None

PLOS authors have the option to publish the peer review history of their article (what does this mean?). If published, this will include your full peer review and any attached files.

Reviewer #1: No

Reviewer #2: Yes: Jose Guerra
---

## [Decision Letter · Decision Letter 1]

21 Jul 2020

Dear Mr. Abbood,

Thank you very much for submitting your manuscript "EventEpi–A Natural Language Processing Framework for Event-Based Surveillance" for consideration at PLOS Computational Biology. As with all papers reviewed by the journal, your manuscript was reviewed by members of the editorial board and by several independent reviewers. The reviewers appreciated the attention to an important topic. Based on the reviews, we are likely to accept this manuscript for publication, providing that you modify the manuscript according to the review recommendations.

Please address reviewer 2's very minor points.

Sincerely,

Benjamin Althouse

Associate Editor

PLOS Computational Biology

Virginia Pitzer

Deputy Editor

PLOS Computational Biology

[LINK]

Please address reviewer 2's very minor points.

Reviewer's Responses to Questions

**Comments to the Authors:**

Reviewer #1: I am satisfied that the authors have addressed the concerns highlighted in the review. I also congratulate them on making the data accessible for future research.

Reviewer #2: Dear Authors,

Thank you for your impressive work in the improvement of the paper, it looks very good now.

I am fully satisfied with your explanations to my questions and the modifications performed to the manuscript.

Please find below some very minor comments and proposed modifications, feel free to consider them or not.

Introduction:

* Line 40: typo, "spent" instead of "spend".

Results:

* Tables 1 and 2: the term "support" is still not very clear, even with the added explanation, maybe you could consider the use of another term such as "sample used".

* The confusion matrices are very good and self-explanatory, I strongly believe they should be part of the main manuscript instead of being in the supplementary material.

S1 Appendix:

* line 554: typo "classification" instead of "classifcation"

Best regards

**Have all data underlying the figures and results presented in the manuscript been provided?**

Reviewer #1: None

Reviewer #2: None

PLOS authors have the option to publish the peer review history of their article (what does this mean?). If published, this will include your full peer review and any attached files.

Reviewer #1: No

Reviewer #2: **Yes: **José Guerra
---

## [Editor Report · Decision Letter 2]

20 Aug 2020

Dear Mr. Abbood,

We are pleased to inform you that your manuscript 'EventEpi–A Natural Language Processing Framework for Event-Based Surveillance' has been provisionally accepted for publication in PLOS Computational Biology.

Best regards,

Benjamin Althouse

Associate Editor

PLOS Computational Biology

Virginia Pitzer

Deputy Editor

PLOS Computational Biology

---

## [Editor Report · Acceptance letter]

27 Oct 2020

PCOMPBIOL-D-19-01790R2 

EventEpi–A Natural Language Processing Framework for Event-Based Surveillance

Dear Dr Abbood,

I am pleased to inform you that your manuscript has been formally accepted for publication in PLOS Computational Biology. Your manuscript is now with our production department and you will be notified of the publication date in due course.

With kind regards,

Matt Lyles
